# Keratinocyte Growth Factor-1 Protects Radioiodine-Induced Salivary Gland Dysfunction in Mice

**DOI:** 10.3390/ijerph17176322

**Published:** 2020-08-31

**Authors:** Jeong Mi Kim, Mi Eun Choi, Seok-Ki Kim, Ji Won Kim, Young-Mo Kim, Jeong-Seok Choi

**Affiliations:** 1Translational Research Center, Inha University, Incheon 22332, Korea; jeongmi77@gmail.com (J.M.K.); cme2417@gmail.com (M.E.C.); hopefuljw@gmail.com (J.W.K.); ymk416@inha.ac.kr (Y.-M.K.); 2Inha Research Institute for Medical Sciences, Inha University College of Medicine, Incheon 22332, Korea; 3Department of Otorhinolaryngology-Head and Neck Surgery, Inha University College of Medicine, Incheon 22332, Korea; 4Department of Nuclear Medicine, National Cancer Center, Goyang 10408, Korea; skkim@ncc.re.kr

**Keywords:** radioiodine, salivary gland, keratinocyte growth factor, anti-apoptosis

## Abstract

Background: Most patients with thyroid cancer suffer from salivary gland (SG) dysfunctions after radioiodine (RI) therapy. We investigated the effects of keratinocyte growth factor (KGF)-1 on RI-induced SG dysfunction in an animal model. Methods: Six C57BL/6 mice were assigned to each of the following groups: treatment naïve control group, RI group, and RI+KGF-1 group. Body and SG weights, salivary flow rates, salivary lag times and changes in 99mTc pertechnetate uptake and excretion were measured, and histologic changes were noted. Amylase activities and epidermal growth factor (EGF) concentrations in saliva were also measured. In addition, TUNEL assays were performed and apoptosis-related protein expressions were assessed. Results: RI-induced reductions in salivary flow rates and increases in salivary lag times observed in the RI group were not observed in RI+KGF-1 group. Mice in RI group had higher HIF1a levels than controls, but HIF1a levels in RI+KGF-1 group were similar to those in control group. Furthermore, mice in RI+KGF-1 group had more mucin stained acini and decreased periductal fibrosis than mice in RI group, and tissue remodeling of many salivary epithelial cells (AQP5) and endothelial cells (CD31) were observed in RI+KGF-1 group. Amylase activity and expression in saliva were greater in RI+KGF-1 group than in RI group, and fewer apoptotic cells were observed in RI+KGF-1 group. Furthermore, BCLxl (anti-apoptotic) expression was higher, and Bax (pro-apoptotic) expression was lower in RI+KGF-1 group than in RI group. Conclusions: Local delivery of KGF-1 might prevent RI-induced SG damage by reducing apoptosis.

## 1. Introduction

Salivary gland (SG) dysfunction is commonly encountered after radioiodine (RI) treatment for differentiated thyroid cancer and causes xerostomia, swallowing difficulties, oral candidiasis, taste loss, and tooth decay. Furthermore, RI-induced SG damage occurs in a dose-dependent manner and when high doses are administered, the damage caused may be permanent and life-threatening. Therefore, many studies have been undertaken to develop means of preventing RI-induced SG dysfunctions, and several treatments based on antioxidants, [1,2] stem cells [3], and bioactive factors [4,5] have been described in the literature.

Keratinocyte growth factor (KGF)-1 (also known as Fibroblast growth factor 7) is a growth factor present during the epithelialization phase of wound healing, and a small signaling molecule that binds to fibroblast growth factor receptor 2b (FGFR2b) [6]. Many studies have shown KGF is potentially useful for protecting and promoting the regeneration of damaged epithelial cells [7,8]. In addition, it has been shown KGF can reduce chemo- or radiotherapy-induced acute and chronic mucositis induced in animal models and that KGF gene transfer can prevent radiation-induced oral mucositis [9,10,11]. KGF-1 also has been reported to have an effect on protecting irradiation-induced salivary dysfunction [4].

Several studies have concluded KGF-1 has regenerative effects on radiation-induced SG damage. However, the effects of KGF-1 on RI-induced SG dysfunction have not been studied. In this study, we investigated the effect of KGF-1 on protecting RI-induced SG dysfunction and sought to elucidate the mechanisms responsible for its radioprotective effect in a mouse model.

## 2. Materials and Methods

### 2.1. Animal Studies

The 18 female mice (C57BL/6; 18–22 g) used in this study were housed in accordance with the guide for the care and use of laboratory animals of the Korean National Cancer Research Center. Mice were classified into three groups (*n* = 6/group): a treatment naïve control group, an RI group (mice were administered phosphate-buffered saline (PBS) plus RI (0.01 mCi/g body weight, ^131^I; New Korea Industrial, Seoul, orally)), and an RI+KGF-1 group (RI as for the RI group and KGF-1). KGF-1 was dissolved in PBS including 0.1% bovine serum albumin (100 ug/KGF-1/1 mL PBS).

After anesthetizing a mouse by injection of drugs (xylazine; 5 mg/kg, i.p. and ketamine; 100 mg/kg, i.p.), a 5-mm neck incision was made to expose submaxillary glands, and 20 µL of KGF-1 or PBS solution was injected directly into the submaxillary gland using a syringe equipped with a 25-gauge needle, 1 h before and immediately after RI administration. After injections, neck wounds were sutured and sterilized. Animals were administered thyroxine (1.5 ug/100 g) and calcium lactate (1%) in drinking water to maintain an euthyroid state up to the end of the experiment (120 days post-RI). All animal experiments were performed in compliance with protocols approved by our Institutional Animal Care and Use Committee (INHA 140220-276).

### 2.2. Measurements of Body and SG Weights, Salivary Flow Rates, and Lag Times

Mice were weighed and injected ketamine and xylazine (i.p.) in sterile water at the end of the experiment, and a freshly prepared solution of pilocarpine in PBS (0.5 mg/mL, 0.01 mL/g body weight) was injected (i.p.) to each mouse. Mice were then placed vertically, and saliva was collected in tubes for 10 min. Salivary lag time was defined as the time taken for saliva to appear after pilocarpine administration. Immediately after saliva collection, mice were euthanized, and submandibular glands were collected; surrounding tissues were removed, and the weights of both glands were measured and recorded.

### 2.3. SPECT Image Analysis

At 120 days post RI, mice were anesthetized and administered technetium pertechnetate i.p. (55.5 MBq, [^99m^Tc] TcO_4_^−^; New Korea Industrial). Whole-body SPECT (single photon emission computed tomography) imaging was started immediately after injecting [^99m^Tc] TcO_4_^−^ and repeated every 5 min until 100 min (NanoSPECT; Bioscan Inc., Washington, DC, USA). A solution of pilocarpine (0.5 mg/mL) in PBS was injected at 0.01 mL/g body weight (i.p.) 1 h after starting SPECT. SPECT images were analyzed in InVivoScope (Bioscan, Washington, DC, USA) and Osirix imaging software (The Osirix Foundation, Geneva, Switzerland) [1].

### 2.4. Histopathology

SG tissues were fixed, embedded, and sectioned. Sections were stained with Masson’s trichrome (MT, Abcam plc, Cambridge, MA, USA) and periodic acid Schiff (PAS, Abcam plc) and examined under a light microscope. Digital images were captured and analyzed.

### 2.5. Immunohistochemistry

Sections of paraffin-embedded tissues were deparaffinized using xylene and then rehydrated using a graded ethanol series. Endogenous peroxidase activities were blocked by treating sections with 0.3% hydrogen peroxide buffer at room temperature (RT) for 20 min. Antigen retrieval was conducted in citrate buffer and then sections were blocked in 10% normal donkey serum for 1 h at RT. They were incubated overnight at 4 °C with rabbit polyclonal antibodies against aquaporin 5 (AQP5) (diluted 1:200; Alomone Labs, Jerusalem, Israel), cluster of differentiation 31 (CD31), and hypoxia-inducible factor 1-alpha (HIF1a) (diluted 1:200; Santa Cruz, CA, USA) using an LSAB kit (Dako, Carpinteria, CA, USA) counterstained with Haemotoxylin and Eosin (H&E), dehydrated, and mounted. The sections were visualized by light microscopy and analyzed.

### 2.6. Detection of Amylase Activity and Expression in Saliva

Saliva samples obtained and were centrifuged (6000 rpm for 15 min) and stored (−70 °C) at 120 days post-RI. Saliva amylase activity was measured using salivary α-amylase assay kit (Salimetrics LLC, State College, PA, USA). Epidermal growth factor (EGF) contents were measured using an enzyme-linked immunosorbent assay (ELISA) kit (Quantikine; R&D systems, Minneapolis, MN, USA). All the experiments were performed according to the protocol. Western blotting was also conducted using an anti-α-amylase antibody (1:1000, Santa Cruz Biotechnology, Santa Cruz, CA, USA) [5].

### 2.7. Detection of Apoptosis and Related Proteins

Apoptosis in submandibular gland tissues was determined using an ApopTag Plus in situ Apoptosis Kit (Chemicon Int., Temecula, CA, USA). Terminal deoxynucleotidyl transferase dUTP nick end labeling (TUNEL)-positive cells were checked at a magnification of ×400, and numbers of TUNEL-positive cells were counted in 10 random fields. To detect apoptosis-related proteins, Western blotting was performed using anti-Bax (Cell Signaling Technology, Beverly, MA, USA) and anti- Bcl-_xl_ (Santa Cruz Biotechnology, Santa Cruz, CA, USA) antibodies.

### 2.8. Western Blotting

Gland samples were homogenized in RIPA lysis buffer containing Complete Mini Protease Inhibitor Cocktail (Roche Diagnostics, Indianapolis, IN, USA), incubated (1 h on ice), and centrifuged (12,000 rpm for 1 h at 4 °C). Boiled proteins (20 µg) were separated and transferred. Membranes were blocked (5% skim milk for 1 h at room temperature), and blots were probed with primary antibodies (anti-Bax and anti-Bcl_XL_; 1:200) and visualized using secondary antibodies (1:5000, Santa Cruz, CA, USA) using an ECL kit (Amersham Pharmacia Biotech, Piscataway, NJ, USA). The relative expression level of the samples was calculated by normalization to the corresponding β-actin (1:5000, Santa Cruz, CA, USA).

### 2.9. Statistical Analysis

Data analysis was conducted using Graph Pad Prism 5 (GraphPad Software Inc., La Jolla, CA, USA). The significances of differences between groups were evaluated using the Kruskal–Wallis test followed by post hoc testing using Dunn’s test. Results are presented as means ± SDs.

## 3. Results

### 3.1. Preventive Effect of KGF-1 on RI Induced Body and SG Weight Losses

No significant intergroup difference in body weights was observed before the experiment. However, at 120 days after RI treatment, body weights were significantly lower in the RI group than in treatment naïve controls (Figure 1A, *p* < 0.05). However, body weights in the RI+KGF-1 and normal groups were similar (*p* > 0.05). SG weights and sizes were also smaller in the RI group than in controls, and as was observed for body weights, SG weights and sizes were similar in the RI+KGF-1 and control group (Figure 1B,C, *p* > 0.05).

### 3.2. Protective Effect of KGF-1 on RI-Induced SG Dysfunction

Salivary flow rates in the RI group were significantly less than in controls at 120 days after RI, but were similar in the RI+KGF-1 and control groups (Figure 2A, both *p* < 0.05, respectively). Salivary lag times were significantly greater in the RI group than in the control and RI+KGF-1 groups (Figure 2B). Furthermore, ^99m^Tc pertechnetate excretion was markedly reduced by RI, but treatment with KGF-1 prevented this reduction (Figure 2C, *p* < 0.05).

### 3.3. Enhanced SG Secretory Function by KGF-1

To determine the effect of KGF-1 on SG secretory function, we compared EGF levels and amylase activities in saliva. Saliva EGF levels were significantly lower in the RI group than in controls but similar in the RI+KGF-1 and control groups (Figure 2D, *p* < 0.05). Amylase activity was significantly lower in the RI group than in controls but similar in RI+KGF-1 and control groups (Figure 2E). Amylase expression was significantly higher in the RI+KGF-1 and controls groups than in the RI group (Figure 2F).

### 3.4. Salivary Cell Protective Effect of KGF-1

Micro-morphological changes in SGs were verified using PAS and MT staining after 120 days RI exposure. PAS staining for mucin density decreased after RI but was higher in the RI+KGF group than in the RI group (*p* < 0.05). MT staining showed RI induced fibrotic changes in the peri-acinoductal area but had less effect in the KGF-1 group (Figure 3A,C, *p* < 0.05). To determine whether RI treatment induced hypoxia, we examined HIF1a expression in SGs. It was found HIF1a was highly expressed in salivary ducts in the RI group but that it was expressed at similar levels in the RI+KGF-1 and control groups (Figure 3B,C, *p* < 0.05). When we examined the cellular protective effects of KGF-1 on salivary epithelial (AQP5) and endothelial (CD31) cells immunohistochemically (Figure 3B,C), we found expressions of AQP5 and CD31 were lower in the RI group than in the RI+KGF-1 and control groups, which suggested KGF-1 protected salivary cells against RI-induced damage (*p* < 0.05).

### 3.5. Anti-Apoptotic Effect of KGF-1 in SGs

Oral RI administration induced apoptosis in SGs (Figure 4). However, apoptotic cell death was found to be reduced by treating mice with KGF-1 before and immediately after RI. TUNEL staining represented a significant increase in apoptosis after RI but markedly less apoptosis after RI+KGF-1 (Figure 4A,B). Furthermore, Western blotting of mouse tissue extracts showed RI treatment increased the expression of Bax (pro-apoptotic) but reduced the expression of Bcl-xl (anti-apoptotic) (Figure 4C). KGF-1 administration significantly inhibited Bax expression and promoted Bcl-xl expression.

## 4. Discussion

Our results suggest KGF-1 might importantly protect against RI-induced salivary gland dysfunction, which is a severe complication of radioiodine ablation of differentiated thyroid cancer. Although the protective effects of KGF-1 have been previously reported in an animal model of radiation injury, ^4^ this is the first study to show KGF-1 might protect SGs from RI.

The β-emitting iodine isotope ^131^I has long been considered to provide the best means of treating thyrotoxicosis and thyroid cancer. ^131^I uptake is almost specific to thyroid tissue, but its off-target accumulation in SG cells causes side effect [12]. The active accumulation of ^131^I in thyroid gland follicular cells is due to the activities of sodium/iodide symporter (NIS), [13] which is expressed in the basolateral membrane of the striated ducts of SGs [14]. The mechanisms of RI-induced SG dysfunction are related to apoptosis, necrosis, and autophagy [15,16,17]. RI causes acute cell death by emitting short path-length (1~2 mm) beta particles and subsequent acute hypoxic damage, which causes the apoptosis and necrosis in various cell types [18,19]. To confirm that RI induced hypoxia, we compared SG expressions of HIF1a in treatment naïve controls and RI exposed mice. We found HIF1a was highly expressed in the salivary ducts of RI exposed mice but that its expression in SG ductal cells of RI+KGF-1 treated mice was at levels similar to those observed in treatment naïve controls.

Choi et al. reported KGF-1 inhibited SG apoptosis caused by irradiation, [4] and Cai et al. found KGF-1 inhibited hypoxia-induced intestinal epithelial cell apoptosis [20], whereas we observed local injections of KGF-1 reduced RI-induced apoptosis in mice. Furthermore, in the present study, KGF-1 treatment reduced the number of apoptotic cells and Bax expression and increased expression of Bcl-xl.

It would appear KGF-1 has the effect on protecting many types of cells from radiation. In the present study, KGF-1 administration to SGs prevented RI damages to acinar, duct, and endothelial cells. Zheng et al. demonstrated that transductal KGF-1 gene delivery to submandibular glands prevented salivary hypofunction by increasing the proliferations of salivary epithelial and endothelial cells [11]. We found local KGF-1 injection protected and promoted SG regeneration and function. In particular, KGF-1 prevented RI-induced body and gland weight reductions and improved salivary flow rates, salivary lag times, and SPECT excretion patterns. Furthermore, EFG and amylase levels were higher in the saliva of RI+KGF-1 treated than in RI treated mice, and histologically, the SG tissues of RI+KGF-1 treated were similar to those of treatment naïve mice.

Growth factors have been used to regenerate SG cells exposed to radiation, [21,22,23] but some growth factors are of limited clinical because of adverse effects, which include carcinogenesis and restriction of cancer therapy [24,25]. In murine xenograft models of head and neck carcinoma, subcutaneously injected KGF-1 did influence tumor growth of chemotherapeutic efficacy [10]. However, due to concerns that systemic KGF-1 delivery might cause malignancies, we administered it by local injection 30 min before and immediately after RI exposure to allow it to mitigate early RI damage.

## 5. Conclusions

Our findings suggest the radio-protective effects of KGF-1 were due to the inhibition of apoptosis, endothelial cell regeneration, the prevention of fibrosis, and the restoration of SG functions. Moreover, our observations indicate SG tissues exposed to RI become hypoxic and undergo apoptosis. Accordingly, we suggest KGF-1 might provide a clinically effective means of alleviating the side effects of RI ablation on SGs and thus of improving the therapeutic efficiencies of RI-based tumor therapies.

## Figures and Tables

**Figure 1 ijerph-17-06322-f001:**
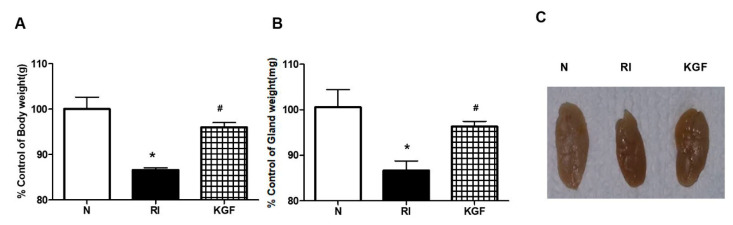
Effect of KGF-1 on salivary gland’s size and weight. (**A**) Body weights in the RI+KGF-1 group were significantly higher than in the RI group. (**B**,**C**) SG weights and sizes were also higher in the RI+KGF-1 group than in the RI group. Results are presented as means ± SDs; *, vs. treatment naïve controls; ^#^, vs. the RI group. * *p* < 0.05, ^#^
*p* < 0.05 (*n* = 6 mice per group) (N; treatment naïve controls, RI; the RI group, KGF; the RI+KGF-1 group).

**Figure 2 ijerph-17-06322-f002:**
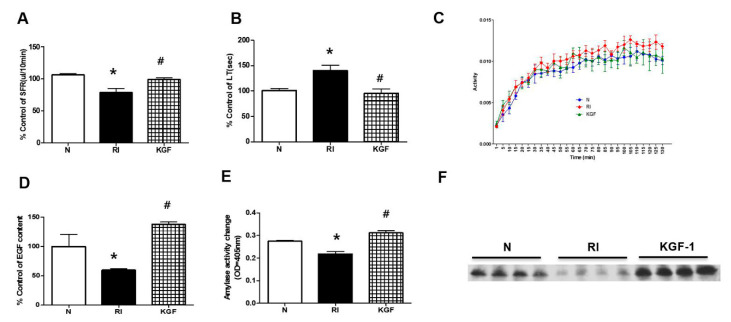
Evaluation of salivary function after RI+KGF-1 treatment. (**A**) Salivary flow rates (SFRs) and (**B**) lag times (LTs). SFRs were greater in the RI+KGF-1 group than in the RI group and LTs were shorter in the RI+KGF-1 group. (**C**) ^99mc^Tc pertechnetate excretion was lower in the RI group than in the other two groups, whereas ^99mc^Tc pertechnetate excretion in the RI+KGF-1 group was similar to that observed in treatment naïve controls. (**D**) EGF levels were higher in the RI+KGF-1 group than in the RI group. (**E**) Salivary amylase activities were higher in the RI+KGF-1 group than in the RI group, and (**F**) amylase expressions were also higher in the RI+KGF-1 group. Results are presented as means ± SDs; *, vs. treatment naïve controls; ^#^, vs. the RI group. * *p* < 0.05, ^#^
*p* < 0.05 (*n* = 6 mice per group) (N; treatment naïve controls, RI; RI group, KGF; RI+KGF-1 group).

**Figure 3 ijerph-17-06322-f003:**
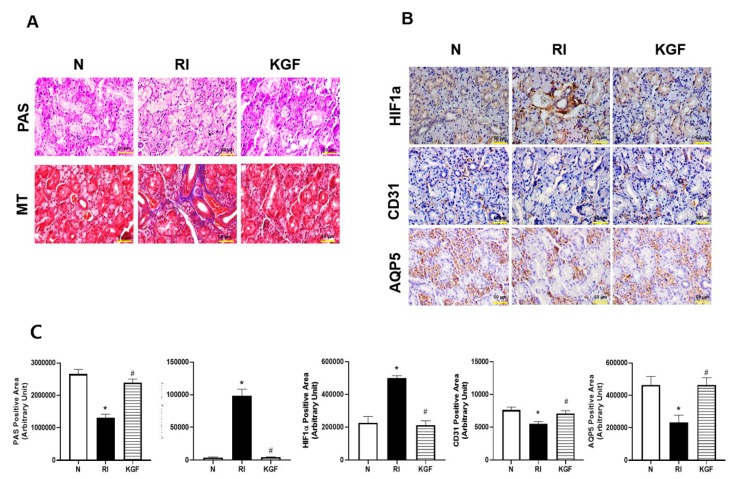
Histological analysis of salivary glands. (**A**) Representative histological images of PAS and MT staining. More mucin-containing acini and less periductal fibrosis were observed in the RI+KGF-1 group than in the RI group. (**B**) Representative immunohistochemical images showing HIF1a, salivary epithelial cells (AQP5 stained) and endothelial cells (CD31 stained). The expression of HIF1a was higher and the expressions of AQP5 and CD31 were lower in the RI group than in treatment naïve controls. HIF1a staining intensities was lesser, and AQP5 and CD31 staining intensities were greater in the RI+KGF-1 group than in the RI group. (N; treatment naïve control group, RI; RI group, KGF; RI+KGF-1 group). Results are presented as means ± SDs; *, vs. treatment naïve controls; ^#^, vs. the RI group. * *p* < 0.05, ^#^
*p* < 0.05 (*n* = 6 mice per group) (N; treatment naïve controls, RI; RI group, KGF; RI+KGF-1 group), Scale bar: 50 μm. (**C**) More mucin-containing acini and less periductal fibrosis were observed in the RI+KGF-1 group than in the RI group. The expression of HIF1a was higher and the expressions of AQP5 and CD31 were lower in the RI group than in treatment naïve controls. HIF1a staining intensities was lesser, and AQP5 and CD31 staining intensities were greater in the RI+KGF-1 group than in the RI group. (N; treatment naïve control group, RI; RI group, KGF; RI+KGF-1 group). Results are presented as means ± SDs; *, vs. treatment naïve controls; ^#^, vs. the RI group. * *p* < 0.05, ^#^
*p* < 0.05 (*n* = 6 mice per group) (N; treatment naïve controls, RI; RI group, KGF; RI+KGF-1 group), Scale bar: 50 μm.

**Figure 4 ijerph-17-06322-f004:**
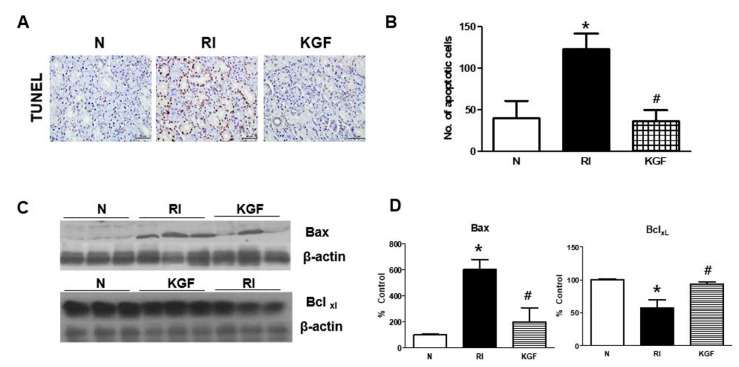
Effects of KGF on RI-induced apoptosis and on apoptosis-related protein expressions. (**A**) Representative TUNEL assay images. Scale bars represent 50 μm. (**B**) Quantitation of TUNEL expression showed it was greater in the RI group than in the RI+KGF-1 group. (**C**) Representative images of the expressions of apoptosis-related protein as determined by Western blotting. (**D**) Bax expression (pro-apoptotic) was higher, and Bcl expression (anti-apoptotic) was lower in the RI group than in the RI+KGF-1 group. β-Actin was used as the loading control. Results are presented as means ± SDs; *, vs. treatment naïve controls; ^#^, vs. the RI group. * *p* < 0.05, ^#^
*p* < 0.05 (*n* = 6 mice per group) (N; treatment naïve control group, RI; RI group, KGF; RI+KGF-1 group).

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
