# Peer review of "Keratinocyte Growth Factor-1 Protects Radioiodine-Induced Salivary Gland Dysfunction in Mice"

_ijerph, 2020, doi:10.3390/ijerph17176322_

Round 1
Reviewer 1 Report
The manuscript titled, “Keratinocyte growth factor-1 (KGF-1) protects radioiodine-2 induced salivary gland dysfunction in mice”, investigates the ability of KGF-1 to ameliorate the effects of thyroid cancer-directed radioiodine treatment on murine salivary glands. The study is simple in design and clearly communicated, and is recommended for publication with the following revisions:
- Figure1 and legend: Please mention all p values including for the RI+KGF-1 group compared to treatment-naive group in the legend to demonstrate KGF-1 impact in getting the endpoint close to treatment-naive group.
- Figure 2 and legend: Were EGF levels higher in RI+KGF groups compared to even N group? What does that mean for adopting KGF as treatment?
- Figure 3: Please insert scale bars for all images.
- Figure 3, panel B: Please quantify histological staining across samples evaluated for HIF1a, CD31 and AQP5, and perform statistics.
Minor revisions:
- Line 30: Please remove “; and (4)” before the work Conslusions.
- Line 31: Please re-phrase “SG damage reducing apoptosis”
- Line 42: Please remove the “s” from “fibroblast growth factors” and capitalize the first letters
- Line 48: Replace “a” with “an”.
- Line 170: Please remove “and”
Author Response
Thank you for your advice on our manuscript, and for the opportunity to revise its contents.
We have corrected items mentioned and have revised the manuscript according to the comments made.

Reviewer 2 Report
Dear Authors,
- Please correct all the typing errors. The number of them is above average –
e.g. line 101-70oC, line 117 – 1hr, etc.
- In the Materials and Methods section – there is missing information regarding the type of secondary antibody, producer and concentration - line 119
- There is missing information regarding the producers of the materials used in the experiment (lines 97, 84, etc- please check the entire text).
- Section 2.1 – did you treat the control group and RI with PBS/0.1% BSA? – if not, please clarify why.
- Fig. 4 – there is no MW information on blots. Bax is ~20 kDa, Bcl-XL - ~38 and B-actin ~45. Pictures show that Bax/Bcl masses are larger than B-actine – can you please explain?
- The most important issue is the novelty of data. Please comment on differences between these data and results presented in the 2017 Oncotarget paper.
E.g. graphs presented at Fig. 4 are similar to the ones at Fig. 4D in the Oncotarget publication.
- Please explain the usage of * and # symbols to indicate p-value – are you sure that all information is correct – e.g. lines 160 and 197 – you marked *p and #p - as <0.05?
- Fig. 8 F – there is no loading control (B-actin) - please provide.
Author Response

(The authors gave the same response as above.)

Round 2
Reviewer 2 Report
Dear Authors,
Thank you very much for provided new version of the manuscript and answers.
I will recommended a paper for publication in IJERPH/MDPI.